# Multi-Stacked Superbuck Converter-Based Single-Switch Charger Integrating Cell Voltage Equalizer for Series-Connected Energy Storage Cells

**Masatoshi Uno \*** , **Qi Xu and Yusuke Sato**

College of Engineering, Ibaraki University, Hitachi 316-8511, Japan; xuqi931218@yahoo.co.jp (Q.X.);
sato.yusuke004@jp.panasonic.com (Y.S.)
**\*** Correspondence: masatoshi.uno.ee@vc.ibaraki.ac.jp

**Abstract:** Voltages of series-connected energy storage cells, such as electric double-layer capacitors (EDLCs) and lithium-ion batteries, need to be equalized to ensure years of safe operation. However, to this end, a voltage equalizer is necessary in addition to a charger, increasing the system complexity and cost. This paper proposes a family of transformerless single-switch integrated chargers that merge a charger and equalizer into a single unit, achieving a simplified system and circuit. Proposed integrated chargers are derived by stacking multiple conventional pulse width modulation (PWM) converters, such as a superbuck converter, that contain two inductors and one energy transfer capacitor. Detailed operation analyses, including an investigation on the impact of component tolerance on voltage equalization performance, are also performed. Experimental charging tests using a 12-W prototype were performed for four EDLC cells. All cells were charged with eliminating voltage imbalance and demonstrating the charging and equalization performance of the proposed integrated charger.

**Keywords:** charger; electric double-layer capacitor (EDLC); superbuck converter; voltage equalizer

## 1. Introduction

Energy storage cells, such as electric double-layer capacitors (EDLCs) and lithium-ion battery cells, are connected in series to form a string to meet the voltage requirements of loads and systems. Individual cell characteristics are not uniform in practical use due to manufacturing tolerance, uneven aging, and ambient temperature conditions. As all cells in a string are charged in series, some cells with higher voltages might be over-charged during a charging process, potentially triggering a hazardous consequence. Even a minor characteristic mismatch eventually results in premature deterioration and reduced capacity, as reported in [1,2]. Thus, cell voltage equalization is mandatory to ensure years of safe operations and to maximize the chargeable/dischargeable energies of cells.

Extensive research and development efforts on cell equalizers have been underway since the advent of EDLCs and lithium-ion batteries. Various types of equalizers, also called balancers, have been proposed and developed. Conventional equalizers based on bidirectional converters, such as pulse width modulation (PWM) converters [3–5] and switched capacitor converters [6–9], transfer energies between neighboring cells or two arbitrary cells to achieve equalization. However, these equalizers are prone to complexity as the switch count necessary is proportional to the number of cells connected in series. With single-input–multi-output converter-based equalizers [10–16], the switch count can be drastically reduced, contributing to the simplified system. Cell equalizers using selection switches, which have been intensively developed for lithium-ion batteries in electric vehicles, can dramatically reduce the passive component count [17–23]. However, these equalizers require numerous bidirectional switches in proportion to the cell count, resulting in increased circuit complexity and cost.

The common drawback of these equalizers is that, in addition to the equalizer, an external converter is necessary for battery regulation. Figure 1a, for example, illustrates an energy storage system based on so-called unregulated bus architecture where the battery is directly connected to a load. This system boasts its simplicity as the battery directly discharges to the load, though the load voltage varies with the battery. Obviously, two separate converters are necessary to perform the two functions of battery charging and cell equalization, thus increasing the system complexity.

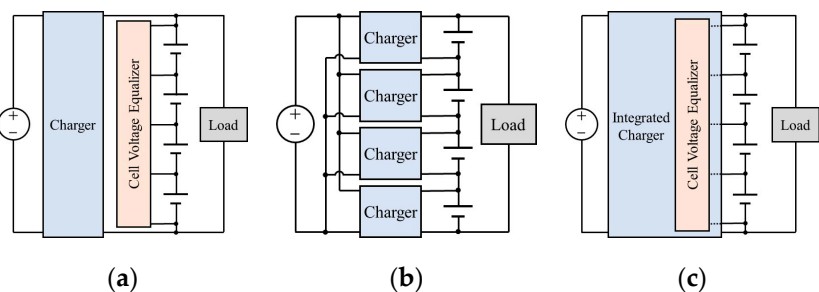

**Figure 1.** Energy storage systems based on (**a**) charger with equalizer, (**b**) individual chargers, and (**c**) integrated charger.

An alternative solution is the use of individual cell chargers, as shown in Figure 1b. All cells can be individually charged to a uniform voltage level by individual chargers. Although cell equalizers are no longer necessary, the charging system apparently tends to be complex as $n$ individual chargers are necessary for $n$ cells connected in series. A multi-output charger based on a multi-winding flyback converter has been proposed [24,25], but the design difficulty of the multi-winding transformer is cited as a top concern—parameters of multiple secondary windings, such as leakage inductance, resistance, and turns ratio, must be strictly matched to achieve an adequate equalization performance.

In previous works [26–28], converters integrating voltage equalizers have been proposed, as shown in Figure 1c. Since a converter (or a charger) and equalizer are integrated through the sharing of some circuit elements, both the system and circuit can be simplified to a greater extent than ordinary systems. Integrated converters would be an appealing topology for low-cost, low-power energy storage systems. However, a transformer is necessary to integrate a converter and equalizer regardless of isolation requirements, increasing the circuit volume and cost. Furthermore, the conventional integrated converters suffer from poor extendibility (or modularity) because redesigning transformers is mandatory in cases where the cell count changes—transformers' turn ratios need to be properly determined based on the number of series-connected cells.

This paper proposes a family of transformerless single-switch integrated chargers that merge a charger and equalizer into a single unit. The proposed integrated chargers can be derived by stacking multiple conventional PWM converters, each having two inductors and one energy transfer capacitor. In addition to the integration, the switch count can be reduced to one, achieving a simplified system, design, and circuit.

This paper is organized as follows. Section 2 presents the circuit derivation, features, and comparison among four integrated charger topologies. Among these is the superbuck converter that is considered best suited for energy storage applications. Detailed operation analyses, including an investigation on the impact of component tolerance on voltage equalization performance, are performed on a superbuck converter-based integrated charger in Section 3. A circuit design for the 12-W prototype is exemplified in Section 4. A dc equivalent circuit of the proposed integrated charger is derived and verified by simulation analysis in Section 5. Finally, Section 6 presents experimental results of the designed 12-W prototype and equalization charging of four EDLCs connected in series.

## 2. Proposed Integrated Chargers

### 2.1. PWM Converters

The proposed integrated chargers are derived from PWM converters as a foundation. PWM converters with two inductors and one energy transfer capacitor, as shown in Figure 2, can be a foundation circuit. A superbuck converter is a step-down converter with low current ripples at both its input and output ports. A single-ended primary inductor converter (SEPIC) and Zeta converters are non-inverting buck-boost converters and are often used as battery chargers for low-power applications. A Ćuk converter is an inverting buck-boost converter.

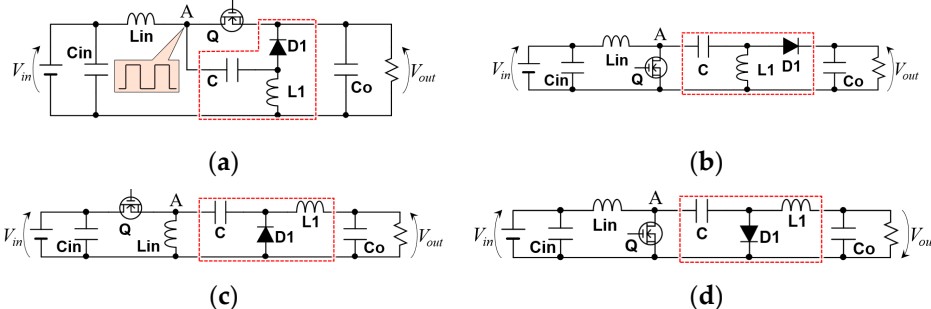

**(a)**                                    **(b)**

**(c)**                                    **(d)**

**Figure 2.** PWM converter with two inductors and one energy transfer capacitor: (**a**) Superbuck converter, (**b**) SEPIC, (**c**) Zeta converter, (**d**) Ćuk converter.

All topologies in Figure 2 contain an energy transfer capacitor followed by an inductor and a diode, as highlighted with dashed lines. Switching nodes (designated as A) produce a square-wave voltage and therefore, the energy transfer capacitors in these topologies behave as an ac-coupling capacitor.

### 2.2. Single-Switch Integrated Chargers

By stacking capacitor−inductor−diode (CLD) circuits on the switching node in respective PWM converters, single-switch integrated chargers can be derived. Derived chargers for four cells connected in series are shown in Figure 3. Smoothing capacitors $C_{o1}$–$C_{o4}$ are connected in parallel with energy storage cells $B_1$–$B_4$. The string voltage, $V_{st}$, is regulated by a constant-voltage (CV) charging scheme, whereas voltages of $C_{o1}$–$C_{o4}$, $V_1$–$V_4$, are automatically equalized even without feedback control, which is detailed in Section 3.

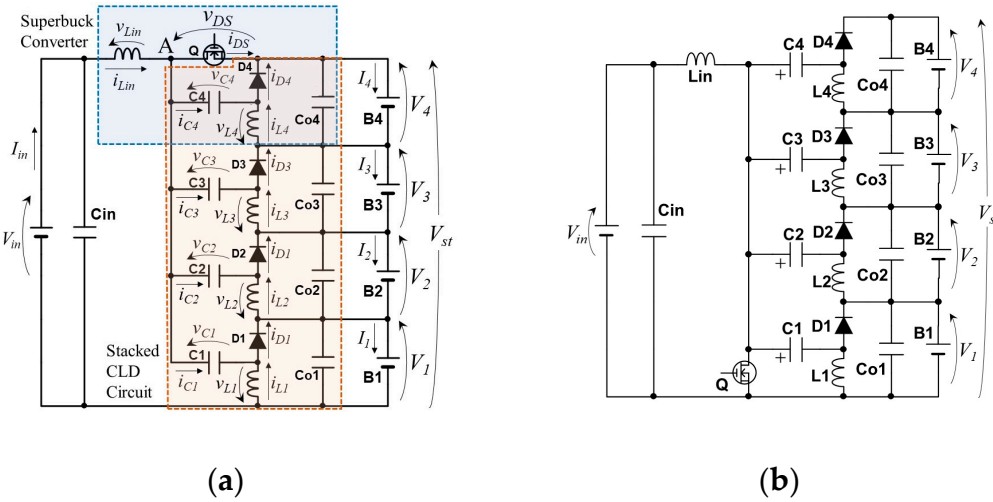

**(a)**                                    **(b)**

**Figure 3.** *Cont.*

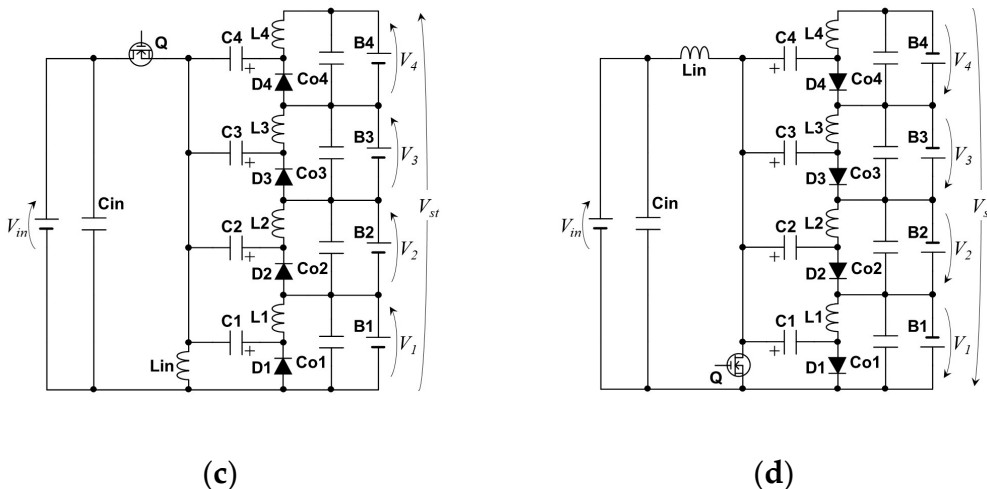

**(c)**　　　　　　　　　　　　　　　　**(d)**

**Figure 3.** Proposed integrated chargers based on (**a**) superbuck converter, (**b**) SEPIC, (**c**) Zeta converter, (**d**) Ćuk converter.

### 2.3. Benefits and Drawbacks

As mentioned in Section 1, the charger and voltage equalizer are integrated into a single unit with a reduction in circuit components, hence realizing a simplified system and circuit. In general, a switch count is a metric of circuit complexity, as each switch requires not only a gate driver circuit but also an auxiliary power supply. Regardless of the cell count, all of the derived integrated chargers contain only one switch, achieving the simplified circuit.

From the viewpoint of power conversion efficiency, the proposed integrated chargers are not a good solution. The integrated chargers contain $n$ diodes (where $n$ is the cell count) and their output voltages are as low as cell voltages. Diode forward voltage drops take significant portions of output voltages, unavoidably lowering power conversion efficiencies. Hence, the proposed integrated chargers would be suitable for low-power applications where circuit simplification and cost reduction are prioritized over efficiency maximization.

The number of cells can be arbitrarily extended by stacking CLD circuits, offering good extendibility. However, voltage conversion ratios depend on topologies and the number of cells (see Section 2.4 for details) and hence, a proper topology needs to be selected when considering applications and requirements.

A charging current for series-connected cells must be properly limited to avoid damage. A constant-current (CC) charging scheme is a straightforward approach to limit a charging current, but obtaining current measurement using a current sensor is necessary. By operating the proposed integrated charger in discontinuous conduction mode (DCM), a charging current can be limited within the desired value even without feedback control, allowing a simplified circuit and reduced cost.

The integrated chargers inherit the advantages and drawbacks of respective foundation circuits listed in Figure 2. In the following subsection, four topologies are compared from various viewpoints.

### 2.4. Comparison

Four integrated chargers are compared from the viewpoint of the DCM boundary condition, current ripple, voltage stress of switches and diodes, and capacitor voltage stress, as shown in Table 1. $V_{in}$ is the input voltage, $d$ is the duty cycle, $V_e$ is the equalized cell voltage ($V_1 = V_2 = \ldots = V_e$), and $n$ is the number of cells in series; equations for the superbuck-based topology will be mathematically derived in Section 3.

**Table 1.** Component values used for prototype.

| Topology | DCM Boundary | Current Ripple | | Voltage Stress of Q and D | Capacitor Voltage $V_{Ck}$ ($k = 1 \ldots n$) |
|---|---|---|---|---|---|
| | | Input | Output | | |
| Superbuck | $\frac{d}{1+d(n-1)}$ | Low | Low | $V_{in} - (n-1)V_e$ | $V_{in} - \sum_{k=1}^{n-1} V_k$ |
| SEPIC | $\frac{d}{1-d}$ | Low | High | $V_{in} + V_e$ | $V_{in} - \sum_{k=1}^{n-1} V_k$ |
| Zeta | $\frac{d}{1-d}$ | High | Low | $V_{in} + V_e$ | $\sum_{k=1}^{n} V_k$ |
| Ćuk | $\frac{d}{1-d}$ | Low | Low | $V_{in} + V_e$ | $V_{in} + \sum_{k=1}^{n} V_k$ |

DCM boundaries are shown and compared in Figure 4. Boundary conditions of the buck-boost converter topologies (i.e., SEPIC, Zeta, and Ćuk converters) are identical, whereas characteristics of the superbuck topology are dependent on $n$. This figure suggests that integrated chargers are prone to operate in continuous conduction mode (CCM) as $V_e/V_{in}$ decreases. The DCM region of the superbuck topology is larger than that of the buck-boost topologies, though its operational region is $nV_e/V_{in} \leq 1.0$. Since the proposed integrated chargers are supposed to operate in DCM, the superbuck topology would be advantageous as long as $nV_e/V_{in} \leq 1.0$ is satisfied.

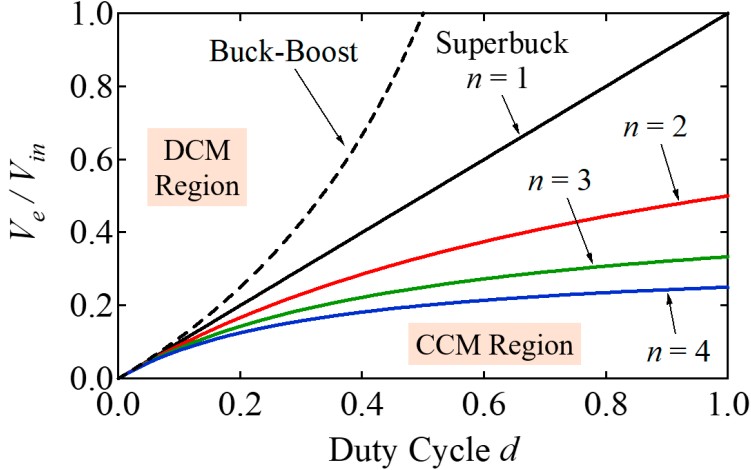

**Figure 4.** DCM boundaries of integrated chargers.

The superbuck-based topology is also superior to other buck-boost topologies in terms of current ripple and voltage stresses. Thus, the following sections focus on the superbuck-based integrated charger.

## 3. Operation Analysis

### 3.1. Voltage Equalization Mechanism

All the energy transfer capacitors, $C_1$–$C_4$, are connected to the switching node generating an ac voltage and hence, the stacked CLD circuits are ac-coupled. Thus, respective CLD circuits, as well as smoothing capacitors and energy storage cells, can be equivalently separated and grounded, as shown in Figure 5. All the CLD circuits and energy storage cells are connected in parallel and are driven by the square-wave voltage generator. This equivalent circuit suggests that a current from the square-wave voltage generator preferentially flows toward the least charged cell(s) with the lowest voltage in the string, and all the cell voltages eventually become uniform as the charging progresses.

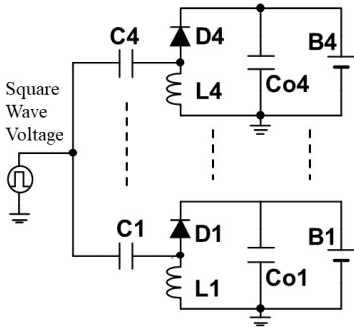

**Figure 5.** Equivalent circuit of stacked CLD circuit.

## 3.2. Operation under Voltage-Imbalanced Condition

This subsection deals with the voltage-imbalanced condition where the voltage of $B_1$, $V_1$, is the lowest in the string. All circuit elements are assumed ideal. Theoretical operation waveforms and current flow directions in DCM are shown in Figures 6 and 7.

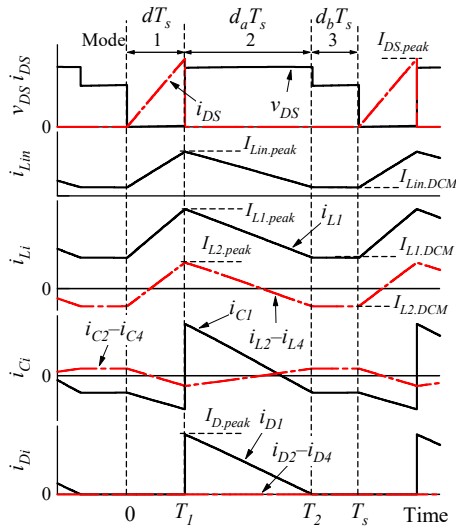

**Figure 6.** Theoretical key operation waveforms under voltage-imbalanced condition.

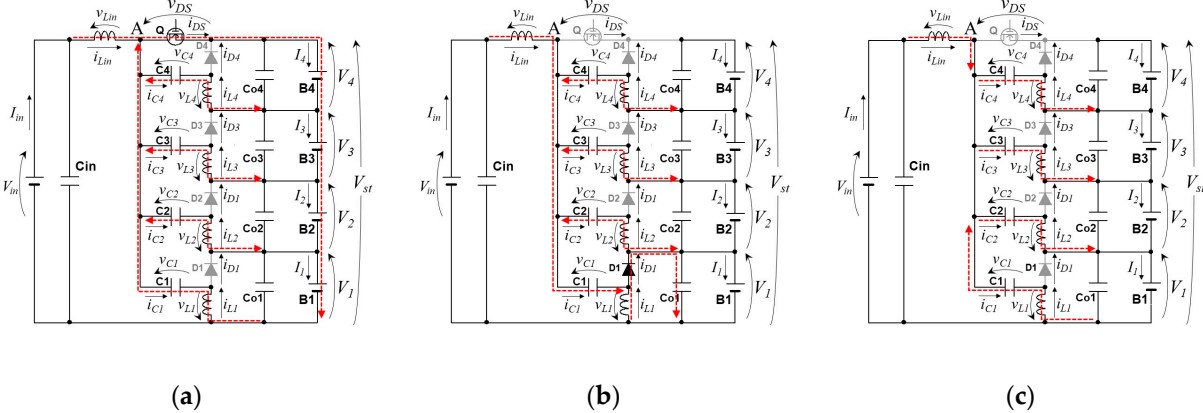

(**a**)　　　　　　　　　　　　(**b**)　　　　　　　　　　　　(**c**)

**Figure 7.** Operation modes under voltage-imbalanced condition in (**a**) Mode 1, (**b**) Mode 2, and (**c**) Mode 3.

Mode 1 ($0 \leq t < T_1$) (Figure 7a): The switch Q is turned on. All inductors are charged, and their currents increase linearly. The voltage applied to $L_{in}$ and $L_i$ ($i = 1 \ldots 4$), $v_{Lin}$ and $v_{Li}$, in Mode 1 are given by

$$v_{Lin} = V_{in} - V_{st} \tag{1}$$

$$v_{Li} = V_{Ci} - \sum_{k=i}^{4} V_k \tag{2}$$

where $V_k$ is the voltage of $B_k$ and $V_{Ci}$ is the voltage of $C_i$ that is expressed as

$$V_{Ci} = V_{in} - \sum_{k=1}^{i-1} V_k \tag{3}$$

This equation can be obtained by assuming average inductor voltages are zero under steady-state conditions. The substitution of (3) into (2) produced

$$v_{Li} = V_{in} - V_{st} \tag{4}$$

Thus, from (1) and (4), $v_{Lin}$ and $v_{Li}$ are identical.
Currents of $L_{in}$ and $L_i$, $i_{Lin}$ and $i_{Li}$, are

$$i_{Lin} = \frac{V_{in} - V_{st}}{L_{in}}t + I_{Lin.DCM} \tag{5}$$

$$i_{Li} = \frac{V_{in} - V_{st}}{L_i}t + I_{Li.DCM} \tag{6}$$

where $I_{Lin.DCM}$ and $I_{Li.DCM}$ are the initial values of $i_{Lin}$ and $i_{Li}$ in Mode 3, as designated in Figure 6. $i_{Lin}$ and $i_{Li}$ peak at the end of this mode as

$$I_{Lin.peak} = \frac{V_{in} - V_{st}}{L_{in}}dT_s + I_{Lin.DCM} \tag{7}$$

$$I_{Li.peak} = \frac{V_{in} - V_{st}}{L_i}dT_s + I_{Li.DCM} \tag{8}$$

The current of Q, $i_{DS}$, is the sum of all the inductor currents of $i_{Lin}$ and $i_{Li}$, and it can be yielded from (5) and (6). The sum of $I_{Lin.DCM}$ and $I_{Li.DCM}$ is zero [see (19)] and therefore,

$$i_{DS} = (V_{in} - V_{st})\left(\frac{1}{L_{in}} + \sum_{k=1}^{4}\frac{1}{L_k}\right)t \tag{9}$$

The peak value of $i_{DS}$, $I_{DS.peak}$, is

$$I_{DS.peak} = I_{Lin.peak} + \sum_{k=1}^{4} I_{Lk.peak} = (V_{in} - V_{st})\left(\frac{1}{L_{in}} + \sum_{k=1}^{4}\frac{1}{L_k}\right)dT_s \tag{10}$$

Mode 2 ($T_1 \leq t < T_2$) (Figure 7b): Q is turned off and $D_1$ starts to conduct. All inductors start discharging. $v_{Lin}$ in Mode 2 is

$$v_{Lin} = V_{in} - V_{C1} - V_1 - V_f = -\left(V_1 + V_f\right) \tag{11}$$

where $V_f$ is the diode forward voltage drop. Meanwhile, $v_{Li}$ in this mode is expressed as

$$\begin{cases} v_{L1} = -\left(V_1 + V_f\right) \\ v_{L2} = -V_{C1} + V_{C2} - V_f \\ v_{L3} = V_2 - V_{C1} + V_{C3} - V_f \\ v_{L4} = V_2 + V_3 - V_{C1} + V_{C4} - V_f \end{cases} \tag{12}$$

Substituting (3) into (12) yields

$$v_{Li} = -\left(V_1 + V_f\right) \tag{13}$$

Similar to Mode 1, $v_{Lin}$ and $v_{Li}$ are identical in this mode.
$i_{Lin}$ and $i_{Li}$ in Mode 2 are expressed as

$$i_{Lin} = -\frac{V_1 + V_f}{L_{in}}(t - T_1) + I_{Lin.peak} \tag{14}$$

$$i_{Li} = -\frac{V_1 + V_f}{L_i}(t - T_1) + I_{Li.peak} \tag{15}$$

The current of $D_1$, $i_{D1}$, is the sum of all the inductor currents, as can be seen from Figure 7b. From (10), (14) and (15),

$$i_{D1} = i_{Lin} + \sum_{k=1}^{4} i_{Lk} = I_{D.peak} - \left(V_1 + V_f\right)\left(\frac{1}{L_{in}} + \sum_{k=1}^{4}\frac{1}{L_k}\right)(t - T_1) \tag{16}$$

where $I_{D.peak}$ is the peak of the diode current and

$$I_{D.peak} = I_{Lin.peak} + \sum_{k=1}^{4} I_{Lk.peak} = I_{DS.peak} \tag{17}$$

This operation mode ends as $i_{D1}$ declines to zero. Hence, from (10) and (17), the mode length can be yielded as

$$d_a T_s = \frac{V_{in} - V_{st}}{V_1 + V_f} d T_s \tag{18}$$

where $d_a$ is the duty cycle of Mode 2.

Mode 3 ($T_2 \leq t < T_s$) (Figure 7c): Both $v_{Lin}$ and $v_{Li}$ are zero and therefore, $i_{Lin}$ and $i_{Li}$ are constant. Kirchhoff's current law at node A yields

$$0 = I_{Lin.DCM} + \sum_{k=1}^{4} I_{Lk.DCM} \tag{19}$$

The voltage conversion ratio under the voltage-imbalanced condition can be yielded from the volt-sec balance of $L_{in}$ from (1) and (11) or $L_1$ of (4) and (13), as

$$V_1 = \frac{d(V_{in} - V_{st})}{d_a} - V_f \tag{20}$$

In summary, the diode $D_1$, which is connected to the least charged cell $B_1$, conducts, whereas the others are off for the entire period. An average current of $D_1$, $I_{D1.ave}$, flows toward $B_1$ as an equalization current is equal to an average current of $i_{L1}$, $I_{L1}$, because an

average current of $C_1$ must be zero under steady-state conditions. $I_{D1.ave}$ and $I_{L1}$ are yielded from (10), (17) and (20), as

$$I_{D1.ave} = I_{L1} = \frac{1}{2}d_a I_{D.peak} = \frac{d^2 T_s (V_{in} - V_{st})^2}{2L_X \left(V_1 + V_f\right)} \tag{21}$$

where $L_X$ is the combined inductance given by

$$\frac{1}{L_X} = \frac{1}{L_{in}} + \sum_{k=1}^{4} \frac{1}{L_k} \tag{22}$$

Meanwhile, an average current of $i_{Lin}$, $I_{Lin}$, must be equal to an average switch current, $I_{Q.ave}$, because average currents of $C_1$–$C_4$ connected to the switching node A are zero under steady-state conditions. $I_{Lin}$ or $I_{Q.ave}$ flowing toward the string is expressed as

$$I_{Lin} = I_{Q.ave} = \frac{1}{2}d I_{DS.peak} = \frac{d^2 T_s (V_{in} - V_{st})}{2L_X} \tag{23}$$

Thus, cells are charged with $I_{Q.ave}$ and $I_{D1.ave}$ under the voltage-imbalanced condition where $B_1$ is the least charged cell. Both $I_{Lin}$ and $I_{L1}$ are dependent on $d_2 T_s$ and $L_x$. Hence, a string charging current ($I_{Lin}$) and equalization current ($I_{L1}$) can be limited within the desired level by properly determining these parameters.

### 3.3. Operation under Voltage-Balanced Condition

Theoretical waveforms and operation modes under the voltage-balanced condition are shown in Figures 8 and 9, respectively. All the inductor currents $i_{Li}$ and the diode current $i_{Di}$ are assumed to be uniform.

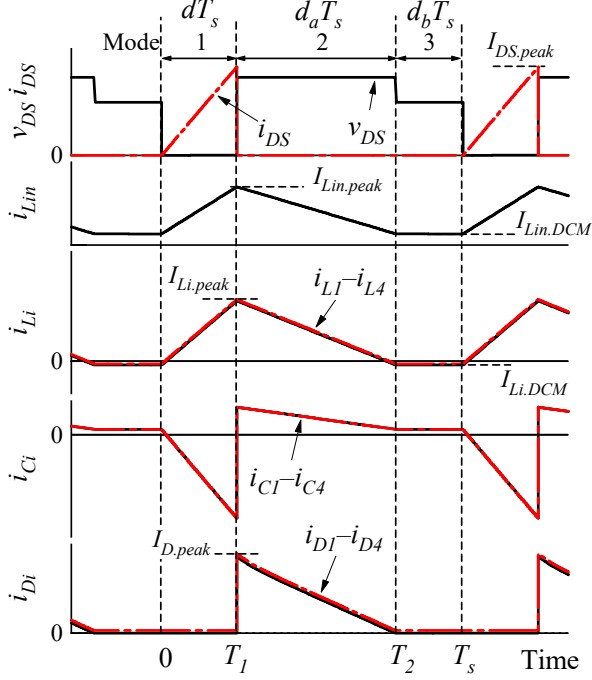

**Figure 8.** Theoretical key operation waveforms under voltage-imbalanced condition.

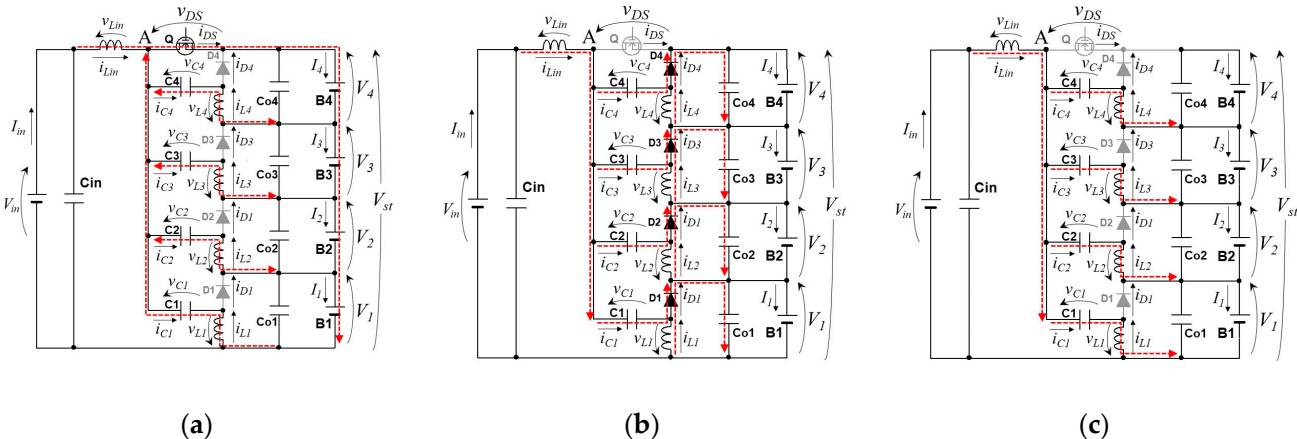

(**a**)                (**b**)              (**c**)

**Figure 9.** Operation modes under voltage-imbalanced condition in (**a**) Mode 1, (**b**) Mode 2, and (**c**) Mode 3.

Mode 1 ($0 \leq t < T_1$) (Figure 9a): Current flow directions under the voltage-balanced condition are identical to those in the voltage-imbalanced condition (see Figure 7a), and therefore, voltages and currents of $L_{in}$, $L_i$, and $D_i$ are expressed identically to those shown in Section 3.2.

Mode 2 ($T_1 \leq t < T_2$) (Figure 9b): $i_{Lin}$ is equally distributed to $C_1$–$C_4$ and flows through $D_1$–$D_4$. $v_{Lin}$ and $v_{Li}$ are

$$v_{Lin} = v_{Li} = -\left(V_i + V_f\right) \tag{24}$$

$i_{Lin}$ and $i_{Li}$ are

$$i_{Lin} = -\frac{V_i + V_f}{L_{in}}(t - T_1) + I_{Lin.peak} \tag{25}$$

$$i_{Li} = -\frac{V_i + V_f}{L_i}(t - T_1) + I_{Li.peak} \tag{26}$$

A current of $D_i$ is the sum of $i_{Li}$ and $i_{Lin}/4$ and therefore,

$$i_{Di} = \frac{i_{Lin}}{4} + i_{Lk} = I_{Di.peak} - \left(V_i + V_f\right)\left(\frac{1}{4L_{in}} + \frac{1}{L_i}\right)(t - T_1) \tag{27}$$

where $I_{Di.peak}$ is the peak of the diode current, calculated as

$$I_{Di.peak} = \frac{I_{Lin.peak}}{4} + I_{Li.peak} = \frac{I_{DS.peak}}{4} \tag{28}$$

This equation reveals that $i_{Di}$ in the voltage-balanced condition is one-fourth of (17). As $i_{Di}$ reaches zero, the operation shifts to the next mode.

Mode 3 ($T_2 \leq t < T_s$) (Figure 9c): Similar to the voltage-imbalanced condition, inductor voltages are zero and inductor currents are constant.

The volt-sec balance of inductors based on (1) or (4) and (24) produces the voltage conversion ratio under the voltage-balanced condition as

$$V_i = \frac{dV_{in} - d_a V_f}{4d + d_a} \tag{29}$$

Substituting $V_{st} = 4V_i$ into (20) leads to (29), indicating that the voltage conversion ratios under the voltage-imbalanced and -balanced conditions are seamless and consistent.

In summary, all diodes conduct in Mode 2 under the voltage-balanced condition. An average current of $D_i$, $I_{Di.ave}$, that is equal to an average of $i_{Li}$, $I_{Li}$, is expressed as

$$I_{Di.ave} = I_{Li} = \frac{1}{2} d_a I_{Di.peak} = \frac{d^2 T_s (V_{in} - V_{st})^2}{8 L_X \left( V_i + V_f \right)} \tag{30}$$

Thus, $I_{Di.ave}$ or $I_{Li}$ under the voltage-balanced condition is a quarter of (21). The comparison between (21) and (30) suggests that the sum of $I_{D1}$–$I_{D4}$ or $I_{L1}$–$I_{L4}$ is independent of whether the cell voltages are balanced. $I_{Lin}$ or $I_{Q.ave}$ are also independent of voltage imbalance as (23) does not contain individual cell voltages, $V_i$.

### 3.4. DCM Boundary

In order for the proposed integrated charger to operate in DCM, Mode 3 must exist. In other words, the sum of the lengths of Modes 1 and 2 must be shorter than a switching period, yielding $d + d_a < 1$. The DCM boundary can be obtained from (18) with the relationship of $d + d_a < 1$, expressed as

$$d < \frac{V_i + V_f}{V_{in} - V_{st} + V_i + V_f} \tag{31}$$

### 3.5. Impact of Component Tolerance on Voltage Equalization Performance

The proposed integrated charger is a single-switch topology and its equalization performance relies on passive components, including capacitors, inductors, and diodes. Hence, component tolerance might impair the voltage equalization performance and its impact should be investigated.

A detailed circuit modeling based on a state−space equation was obtained for the circuit using two cells connected in series (see Figure 10):

$$\begin{cases} \frac{dx}{dt} = Ax + Bu \\ \quad y = Cx \end{cases} \tag{32}$$

where $x$ is the state variable vector, $u$ is the input vector, and $y$ is the output vector;

$$x = \begin{bmatrix} i_{Lin} & i_{L1} & i_{L2} & v_{C1} & v_{C2} & v_1 & v_2 \end{bmatrix}^T \tag{33}$$

$$u = \begin{bmatrix} V_{in} & V_{D1} & V_{D2} & I_1 & I_2 \end{bmatrix}^T \tag{34}$$

$$y = \begin{bmatrix} v_1 & v_2 \end{bmatrix}^T \tag{35}$$

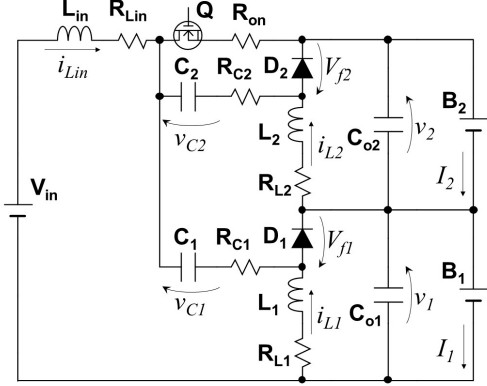

**Figure 10.** Integrated charger for state−space modeling.

Coefficient matrices *A* and *B* are expressed as

$$
A = \begin{bmatrix}
\frac{(R_{Lin}+dR_{on})(R_{C1}+R_{C2})+(1-d)R_{C1}R_{C2}}{L_{in}} & \frac{dR_{on}}{L_{in}} & \frac{dR_{on}}{L_{in}} & \frac{R_{C2}(1-d)}{L_{in}(R_{C1}+R_{C2})} & \frac{R_{C1}(1-d)}{L_{in}(R_{C1}+R_{C2})} & \frac{1}{L_{in}} & \frac{R_{C1}+dR_{C2}}{L_{in}(R_{C1}+R_{C2})} \\
\frac{dR_{on}}{L_1} & \frac{dR_{on}+dR_{C1}+R_{L1}}{L_1} & \frac{dR_{on}}{L_1} & -\frac{d}{L_1} & 0 & \frac{1}{L_1} & \frac{d}{L_1} \\
\frac{dR_{on}}{L_2} & \frac{dR_{on}}{L_2} & \frac{dR_{on}+dR_{C2}+R_{L2}}{L_2} & 0 & -\frac{d}{L_2} & 0 & \frac{1}{L_2} \\
\frac{-R_{C2}(1-d)}{C_1(R_{C1}+R_{C2})} & \frac{d}{C_1} & 0 & \frac{1-d}{C_1(R_{C1}+R_{C2})} & \frac{-(1-d)}{C_1(R_{C1}+R_{C2})} & 0 & \frac{-(1-d)}{C_1(R_{C1}+R_{C2})} \\
\frac{-R_{C1}(1-d)}{C_2(R_{C1}+R_{C2})} & 0 & \frac{d}{C_2} & \frac{-(1-d)}{C_2(R_{C1}+R_{C2})} & \frac{1-d}{C_2(R_{C1}+R_{C2})} & 0 & \frac{1-d}{C_2(R_{C1}+R_{C2})} \\
-\frac{1}{C_{o1}} & -\frac{1}{C_{o1}} & 0 & 0 & 0 & 0 & 0 \\
-\frac{R_{C1}+dR_{C2}}{C_{o2}(R_{C1}+R_{C2})} & -\frac{d}{C_{o2}} & -\frac{1}{C_{o2}} & \frac{-(1-d)}{C_{o2}(R_{C1}+R_{C2})} & \frac{1-d}{C_{o2}(R_{C1}+R_{C2})} & 0 & \frac{1-d}{C_{o2}(R_{C1}+R_{C2})}
\end{bmatrix} \quad (36)
$$

$$
B = \begin{bmatrix}
\frac{-1}{L_{in}} & \frac{R_{C2}(1-d)}{L_{in}(R_{C1}+R_{C2})} & \frac{R_{C1}(1-d)}{L_{in}(R_{C1}+R_{C2})} & 0 & 0 \\
0 & \frac{1-d}{L_1} & 0 & 0 & 0 \\
0 & 0 & \frac{1-d}{L_2} & 0 & 0 \\
0 & \frac{1-d}{C_1(R_{C1}+R_{C2})} & \frac{-(1-d)}{C_1(R_{C1}+R_{C2})} & 0 & 0 \\
0 & \frac{-(1-d)}{C_2(R_{C1}+R_{C2})} & \frac{1-d}{C_2(R_{C1}+R_{C2})} & 0 & 0 \\
0 & 0 & 0 & \frac{1}{C_{o1}} & 0 \\
0 & \frac{-(1-d)}{C_{o2}(R_{C1}+R_{C2})} & \frac{1-d}{C_{o2}(R_{C1}+R_{C2})} & 0 & \frac{1}{C_{o2}}
\end{bmatrix} \quad (37)
$$

Based on the derived state−space equation, the sensitivity analysis for the voltage error of $(v_1 - v_2)/v_1$ was performed. The percentage impacts of $\pm 10\%$ component tolerance were investigated based on both the state−space equation and the simulation analysis. Results are shown in the form of a tornado diagram in Figure 11 and typical values and analysis conditions are listed in the inset. The theoretical and simulation results were in good agreement, verifying the derived state−space equation. The results indicated the diode forward voltage drops of $V_{f1}$ and $V_{f2}$ were the largest source of the voltage error. If $V_{f1}$ increased by 10%, for example, the voltage error would be $-1.8\%$. The impact of the component tolerance of inductances and capacitances was minor because cell voltages are theoretically independent of these parameters, as indicated by (20) and (29). In summary, component tolerances showed a minor impact on the voltage error, suggesting that cell voltages can be equalized well by the proposed integrated charger even without carefully screening circuit components.

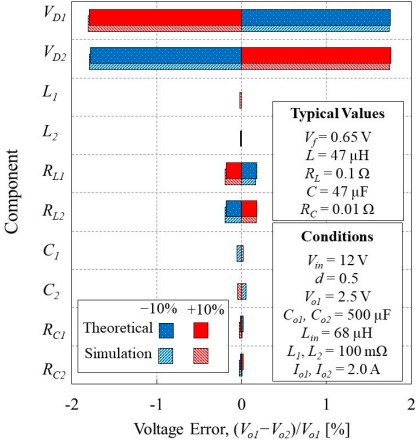

**Figure 11.** Percentage impact of $\pm 10\%$ change in component values on voltage error.

## 4. Design Example

The proposed integrated charger for four EDLCs and low-power ac adaptors with $V_{in}$ = 19.5 V is exemplified in this section. The design target is $P_{in}$ = 12 W (i.e., $I_{Lin} < 0.62$ A), $V_{st} > 6$ V, $V_i$ = 1.2–2.5 V, and $f_s$ = 50 kHz ($T_s$ = 20 μs).

To ensure DCM operations, (31) must be satisfied, yielding

$$\frac{1.2V + 0.3V}{19.5V - 6V + 1.2V + 0.3V} > d = 0.1 \tag{38}$$

The combined inductance $L_x$ can be determined from (23), as

$$I_{Lx} = \frac{0.1^2 \times 20\mu s \times (19.5V - 6V)}{2 \times 0.62A} \approx 2.2\mu H \tag{39}$$

Given $L_{in} = L_i$ for the sake of design simplicity, (22) yields

$$L_{in} = L_i = 5 \times 2.2\mu H = 11\mu H \rightarrow 10\mu H \tag{40}$$

$C_i$ needs to be sufficiently large to ensure that the resonance between $L_i$ and $C_i$ does not occur. To this end, $C_i$ is determined such that the resonant frequency is lower than one-fifth of $f_s$.

$$\frac{f_s}{5} = \frac{1}{2\pi\sqrt{L_i C_i}} \rightarrow C_i = 36\mu F \tag{41}$$

## 5. DC Equivalent Circuit and Its Simulation Results

### 5.1. Derivation of DC Equivalent Circuit

The operation analysis in Sections 3.2 and 3.3 derived the average inductor currents and revealed that the sum of $I_{L1}$–$I_{L4}$ is independent of whether the cell voltages are balanced or imbalanced. A dc equivalent circuit of the superbuck-based integrated charger can be derived by expressing inductors as a constant current source, as shown in Figure 12. Current sources of $I_{Lin}$ and $I_{L1}$–$I_{L4}$ obey (23) and (30), respectively. To generate a seamless transition between (21) and (30), $I_{L1}$–$I_{L4}$ are connected in parallel through an ideal multi-winding transformer. Under voltage-balanced conditions, $I_{L1}$–$I_{L4}$ flow toward $B_1$–$B_4$ through their respective diodes. On the other hand, under voltage-imbalanced conditions, all of $I_{L1}$–$I_{L4}$ go to the least charged cell via the multi-winding transformer.

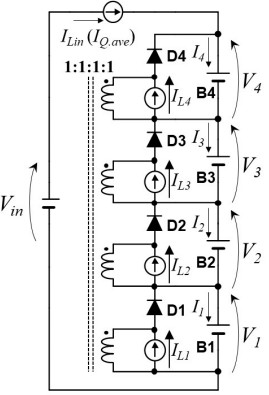

**Figure 12.** DC equivalent circuit of integrated charger for four cells in series.

### 5.2. Simulation-Based Equalization

A simulation-based equalization test using the derived dc equivalent circuit was performed for four cells connected in series. 400-F capacitors were used as cells. $I_{Lin}$ and $I_{L1}$–$I_{L4}$ were programmed to obey (23) and (30), respectively, and component values for an experimental prototype (see Table 2) at a switching frequency of 50 kHz with $d = 0.1$ were used for the simulation. The series-connected cells were charged to be a CV of 10.0 V (2.5 V/cell).

**Table 2.** Component values used for prototype.

| Component | Value, Part Number |
|---|---|
| Q | N-Ch MOSFET, ZXMN4A06GTA, $R_{on}$ = 75 mΩ |
| $L_{in}$, $L_1$–$L_4$ | 10 μH, 33 mΩ |
| $C_{in}$ | Aluminum Electrolytic Capacitor, 330 μF |
| $C_1$–$C_4$ | Ceramic Capacitor, 36 μF |
| $D_1$–$D_4$ | Schottky Barrier Diode, SL44, $V_f$ = 0.35 V |
| $C_{o1}$–$C_{o4}$ | Ceramic Capacitor, 300 μF |
| Gate Driver | L6741 |

The charging profiles are shown in Figure 13. The input current $I_{in}$ steadily decreased as $V_{st}$ increased [see (23)]. Equalization currents, or $I_{L1}$–$I_{L4}$, flowed toward cells, but their magnitudes were dependent on cell voltages. At the beginning of the charging, $B_4$ received the largest equalization current in the form of $I_{L4}$, and $V_4$ increased faster than others. The voltage imbalance was gradually eliminated as the charging progressed, and all the cells were uniformly charged to 2.5 V after $V_{st}$ reached the CV charging level of 10.0 V.

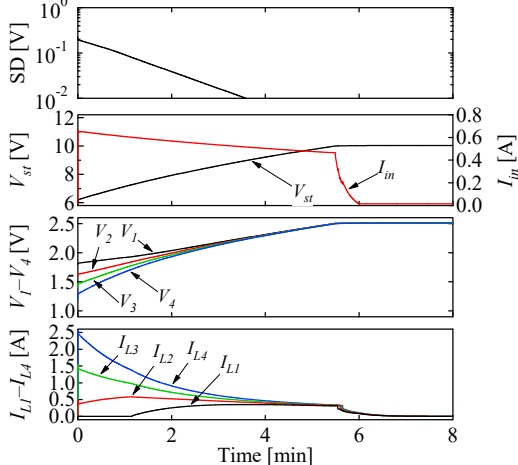

**Figure 13.** Charging profiles of dc equivalent circuit.

## 6. Experimental Results

### 6.1. Prototype and Its Characteristics

A 12-W prototype for four cells connected in series was built, as shown in Figure 14. Circuit elements used for the prototype are listed in Table 2. The prototype was driven at $f_s$ = 50 kHz with $V_{in}$ = 19.5 V and fixed at $d$ = 0.2 to ensure the DCM operations.

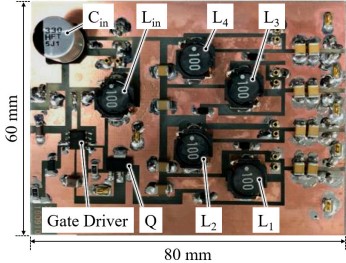

**Figure 14.** Photograph of 12-W prototype.

### 6.2. Characteristics of Integrated Charger Alone

Output characteristics of the prototype were measured using the experimental setup shown in Figure 15. Instead of using actual EDLC cells, two voltage loads were used to emulate voltage-imbalanced conditions. The upper load was a CV load, while the lower one was a variable voltage (VV) load to emulate low-voltage cells. For example, Tap 1

emulates the imbalanced case where $V_1$ is the lowest—Tap 1 emulates the operation modes in Figure 7. The CV load is set at 7.5 V (2.5 V/cell), and the VV load is swept in the range of 1.2–2.5 V. Tap 2, on the other hand, emulates the case in which both $V_1$ and $V_2$ are the lowest. The CV load voltage is fixed at 5.0 V (2.5 V/cell) while sweeping the VV load in the range of 2.4–5.0 V (1.2–2.5 V/cell). The operation modes under the voltage-balanced condition (see Figure 9) can be emulated by selecting Tap 4, through which the CV load is short-circuited, and the entire string is connected to the VV load.

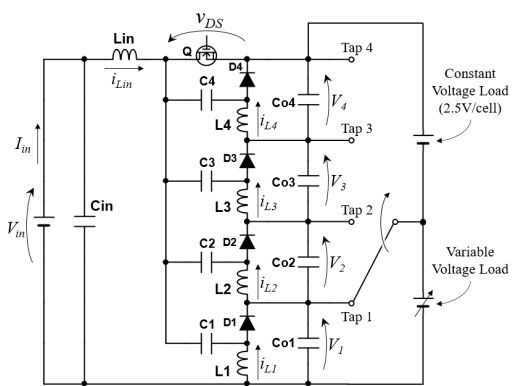

**Figure 15.** Experimental setup for characteristic measurement.

The measured key operation waveforms when $V_1$ = 1.2 V are shown in Figure 16, where $v_{GS}$ is the gate-source voltage. Oscillations in $v_{DS}$ were due to the parasitic capacitance of the MOSFET. When Tap 1 was selected (Figure 16a), the average of $i_{L1}$ was substantial, whereas those of $i_{L2}$–$i_{L4}$ were uniform and zero. These measured waveforms agreed well with the theoretical ones shown in Figure 6. In the cases where Taps 2 and 3 were selected (see Figure 16b,c), $i_{L1}$ and $i_{L2}$ were uniform and their averages were greater than zero, whereas the averages of the other inductor currents were zero. When Tap 4 was selected to emulate the voltage-balanced condition, $i_{L1}$–$i_{L4}$ were uniform.

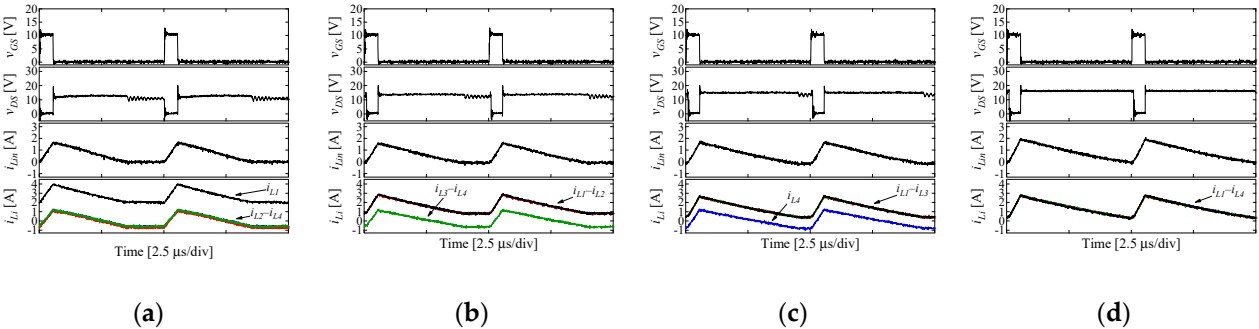

**Figure 16.** Measured waveforms with (**a**) tap 1, (**b**) tap 2, (**c**) tap 3, and (**d**) tap 4.

Measured power conversion efficiencies as a function of $V_1$ are shown in Figure 17—$V_1$ corresponds to the least charged voltage in practical use. Measured efficiencies monotonically increased with $V_1$ as the portion of the output voltage taken by the diode voltage drops decreased. Since the output voltage was low (<2.6 V), diode forward voltage drops took a significant portion of the output voltage $V_1$, resulting in poor power conversion efficiencies of <90%. The efficiencies varied depending on the selected tap, probably because the number of conducting diodes differed depending on whether cell voltages were balanced—only one diode conducts in the case of tap 1 (see Figure 7b), whereas all diodes conduct in the case of tap 4 (Figure 9b). These results suggest that the proposed integrated charger is not suited for high-power applications where efficiency maximization is prioritized over the circuit simplification, as mentioned in Section 2.3.

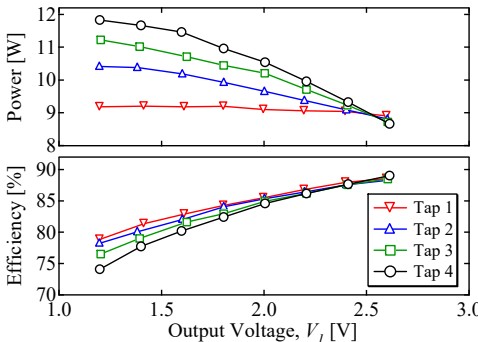

**Figure 17.** Experimental setup for fundamental characteristic measurement.

### 6.3. Charging Test for EDLCs

Four EDLCs, each with a capacitance of 400 F at a rated charging voltage of 2.5 V, were charged using the prototype with $d = 0.1$. Initial cell voltages were imbalanced between 1.2 and 1.8 V. The CV charging voltage level was set to be 10.0 V (2.5 V/cell).

The resultant charging profiles are shown in Figure 18a. The voltage imbalance was gradually eliminated as the charging progressed, and all the cell voltages were satisfactorily equalized in the CV charging period. The standard deviation (SD) of cell voltages decreased to values as low as 11 mV at the end of the charging test, demonstrating the equalization performance of the proposed integrated charger.

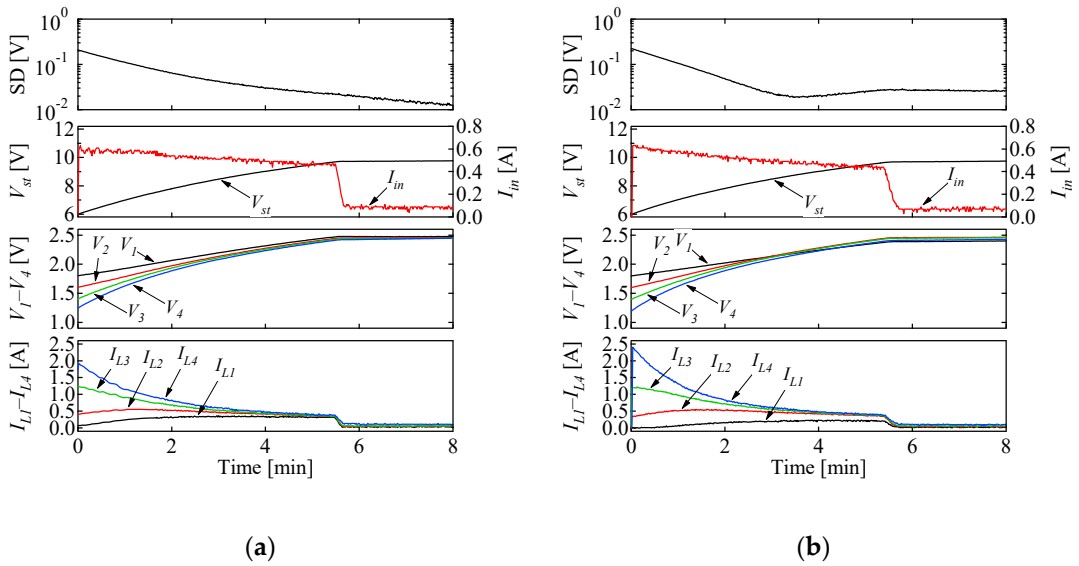

(**a**)            (**b**)

**Figure 18.** Resultant charging profiles of EDLCs with (**a**) uniform circuit elements and (**b**) mismatched circuit elements.

Only the diode $D_1$ was replaced with the one with $V_f = 0.4$ V in order to investigate the impact of component tolerance on the voltage equalization performance—The $V_f$ of other diodes was 0.35 V, as shown in Table 2. The results of the charging test are shown in Figure 18b. Even with the mismatched $V_f$ of $D_1$, the voltage imbalance adequately disappeared, and all the cells were uniformly charged to 2.5 V in the CV charging period. The SD at the end of the CV charging was 25 mV, which was slightly larger than the case shown in Figure 18a. Although slightly increased, the results demonstrated the minor impact of the component tolerance on the equalization performance.

The experimental results agreed very well with the simulation results shown in Figure 13, verifying the derived dc equivalent circuit (see Figure 12).

## 7. Conclusions

A family of single-switch transformerless integrated chargers have been proposed in this paper. The proposed chargers can be derived from stacking multiple PWM converters that contain two inductors and one energy transfer capacitor. The superbuck converter-based topology, which is considered best suited for energy storage applications, was analyzed under voltage-balanced and -imbalanced conditions.

The impact of component tolerance on voltage equalization performance of the integrated charger for two cells was analyzed based on the state−space modeling. The analytical results revealed that the mismatch in diode forward voltage drop was the most influential factor, but ±10% tolerance results in merely 1.8% voltage error, suggesting no significant impact due to component tolerance on the voltage equalization performance.

The 12-W prototype for four cells connected in series was designed and built for the experimental verification tests. Series-connected EDLCs were charged in voltage-imbalanced conditions. All the cells were charged to uniform voltage levels while eliminating voltage imbalance, demonstrating the voltage equalization performance of the proposed integrated charger.

**Author Contributions:** Conceptualization, M.U.; methodology, M.U. and Q.X.; software, Q.X.; validation, Q.X. and Y.S.; formal analysis, M.U. and Q.X.; writing—original draft preparation, M.U.; writing—review and editing, M.U.; supervision, M.U. and Y.S.; project administration, M.U. All authors have read and agreed to the published version of the manuscript.

**Funding:** This research received no external funding.

**Institutional Review Board Statement:** Not applicable.

**Informed Consent Statement:** Not applicable.

**Data Availability Statement:** Not applicable.

**Conflicts of Interest:** The authors declare no conflict of interest.

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
