# Peer review of "Multi-Stacked Superbuck Converter-Based Single-Switch Charger Integrating Cell Voltage Equalizer for Series-Connected Energy Storage Cells"

_energies, doi:10.3390/en15103619_

Round 1

Reviewer 1 Report

This manuscript proposed different topologies for voltage equalization of series-connected energy storage elements such as electric double-layer capacitors (EDLCs) and integrate with charger in a single unit in order to simplify the circuit and system. The superbuck converter based topology concluded as the best for energy storage applications among other proposed ones. The manuscript includes a good literature survey, analysis, simulation, and experimental results, and comparison which results in a complete paper. I would suggest checking the manuscript for a few typo mistakes.

Author Response

Dear Reviewers

We are very grateful to you for your thoughtful and helpful review of the manuscript. Your comments and suggestions have been incorporated as appropriate into the revised manuscript. The revised parts are highlighted with green in the revised manuscript. The responses to reviewers’ comments are noted below.

Reviewer #1

Comment #1

This manuscript proposed different topologies for voltage equalization of series-connected energy storage elements such as electric double-layer capacitors (EDLCs) and integrate with charger in a single unit in order to simplify the circuit and system. The superbuck converter based topology concluded as the best for energy storage applications among other proposed ones. The manuscript includes a good literature survey, analysis, simulation, and experimental results, and comparison which results in a complete paper. I would suggest checking the manuscript for a few typo mistakes.

Response

  Thank you so much for your kind and helpful review. We have reviewed the paper again and have carefully modified typos before resubmission.

Reviewer 2 Report

This paper presents a multi-stacked superbuck converter that is capable of self-equalizing the voltage of series-connected energy storage cells. The paper is well written, explains the analytical analysis in detail, and is validated by experimental results. I think the author misplaced Figure 1 a and b. furthermore, figure 17 needs more explanation.

Author Response

Dear Reviewers

We are very grateful to you for your thoughtful and helpful review of the manuscript. Your comments and suggestions have been incorporated as appropriate into the revised manuscript. The revised parts are highlighted with green in the revised manuscript. The responses to reviewers’ comments are noted below.

Reviewer #2

Comment #1

This paper presents a multi-stacked superbuck converter that is capable of self-equalizing the voltage of series-connected energy storage cells. The paper is well written, explains the analytical analysis in detail, and is validated by experimental results. I think the author misplaced Figure 1(a) and (b). furthermore, figure 17 needs more explanation in the revised manuscript (Section 6.2 in page 15 line 337–344).

Response

  Thank you so much for pointing out. We have modified Figs. 1(a) and (b). As for Fig. 17, we have reconsidered the discussion, as below (page 15 line 338–348).

Measured power conversion efficiencies as a function of V1 are shown in Fig. 17—V1 corresponds to the least charged voltage in practical use. Measured efficiencies mono-tonically increased with V1 as the potion of the output voltage taken by diode voltage drops decreased. Since the output voltage was low (< 2.6 V), diode forward voltage drops took a significant portion of the output voltage V1, resulting in poor power conversion efficiencies of < 90%. The efficiencies varied depending on the selected tap, probably because the number of conducting diodes differed depending on whether cell voltages were balanced—only one diode conducts in the case of tap 1 [see Fig. 7(b)], whereas all diodes conduct in the case of tap 4 [Fig. 9(b)]. These results suggest that the proposed integrated charger is unsuitable for high-power applications where efficiency maximization is prioritized over the circuit simplification, as mentioned in Section 2.3.

Reviewer 3 Report

In the present manuscript “Multi-Stacked Superbuck Converters-Based Single-Switch Charger Integrating Cell Voltage Equalizer for Series-Connected Energy Storage Cells”, Masatoshi Uno and colleagues rationally proposed a family of transformerlesss single-switch integrated chargers. The authors systematically compared four topologies with simulation and experiments. Overall, I think that the manuscript is well-structured, well-written, and the data are of potential relevance on a current topic of interest.

I have little suggestions to improve the quality of paper.

1) 'PWM' should be mentioned in full form at line 14 and 35 before the abbreviation is used later.

2) 'ID.peak' at line 228 should be 'IDi.peak'.

Author Response

Dear Reviewers

We are very grateful to you for your thoughtful and helpful review of the manuscript. Your comments and suggestions have been incorporated as appropriate into the revised manuscript. The revised parts are highlighted with green in the revised manuscript. The responses to reviewers’ comments are noted below.

Reviewer #3

Comment

In the present manuscript “Multi-Stacked Superbuck Converters-Based Single-Switch Charger Integrating Cell Voltage Equalizer for Series-Connected Energy Storage Cells”, Masatoshi Uno and colleagues rationally proposed a family of transformerlesss single-switch integrated chargers. The authors systematically compared four topologies with simulation and experiments. Overall, I think that the manuscript is well-structured, well-written, and the data are of potential relevance on a current topic of interest.

I have little suggestions to improve the quality of paper.

Comment #1

‘PWM’ should be mentioned in full form at line 14 and 35 before the abbreviation is used later.

Response

We appreciate your suggestions. We have added the full form of PWM in the abstract (line 14) and the introduction (line 37).

Comment #2

2) ‘ID.peak’ at line 228 should be ‘IDi.peak’.

Response

  Thank you so much for pointing out. We should have carefully checked the manuscript. We have corrected as ‘IDi.peak’ in line 230. And we also have checked and rechecked the manuscript before resubmission.

Reviewer 4 Report

The paper presents a family of transformerless single-switch integrated chargers that merge a charger and equalizer into a single unit.

Analytical description, simulation and experimentally results are presented.

I consider the paper as a complete presentation of the proposed solution.

I suggest to add in the first sentence of the Abstract:

Voltages of series-connected energy storage cells, such as electric double-layer capacitors (EDLCs) and lithium ion battery cells ...

Author Response

Dear Reviewers

We are very grateful to you for your thoughtful and helpful review of the manuscript. Your comments and suggestions have been incorporated as appropriate into the revised manuscript. The revised parts are highlighted with green in the revised manuscript. The responses to reviewers’ comments are noted below.

Reviewer #4

Comment

The paper presents a family of transformerless single-switch integrated chargers that merge a charger and equalizer into a single unit. Analytical description, simulation and experimentally results are presented. I consider the paper as a complete presentation of the proposed solution.

I suggest to add in the first sentence of the Abstract: Voltages of series-connected energy storage cells, such as electric double-layer capacitors (EDLCs) and lithium ion battery cells ...

Response

  Thank you so much for the productive suggestion. We have added the description of lithium-ion batteries in the abstract (line 10 in page 1).
